
# Characteristics of debris flows recorded in the Shenmu area of central Taiwan between 2004 and 2021

Yi-Min Huang[1]

[1]Department of Civil Engineering, Feng Chia University, Taichung, 407, Taiwan, R.O.C.

*Correspondence to*: Yi-Min Huang (ninerh@mail.fcu.edu.tw)

**Abstract.** The data of debris-flow events between 2004 and 2001 in the Shebmu area Taiwan were presented and discussed in this paper. Total of 20 debris flows were observed in this time interval. The monitored data include rainfall, flow velocity, soil moisture, and ground surface vibrations. Debris flows in the Shenmu area usually occurred in the Aiyuzi Stream during
the rainy season, May to September, and about once per year after 2009. The measured rainfalls from separate monitoring stations were compared and the rainfall thresholds of the Shenmu area were analysed using the rainfall intensity ($I$), the accumulated rainfall ($R$), and durations ($D$). The dataset and the rainfall thresholds of the Shenmu area permits the comparison with other monitored catchments.

## 1 Introduction

The debris flows have become a common disaster in Taiwan in the past two decades (Huang et al., 2013; Huang et al., 2016; Huang et al., 2017). To understand the characteristics, especially the triggering factors, of debris flows, devices and monitoring stations had been installed and established at the areas prone to the debris flows. It is difficult for debris-flow research to obtain the full-scale experimental data that can represent the in-situ conditions. Instead, the observation data from these monitoring stations in Taiwan provides valuable information for debris-flow characteristic studies (Huang et al., 2013;
Hürlimann et al, 2019). In a potential area, the debris flows usually occur in a low frequency and are considered as uncommon events when comparing to other natural hazards, e.g., the heavy rainfalls. But at some locations, the occurrence frequency of debris flow is higher than other locations (Marchi et al, 2021). The Shenmu area in Taiwan is the one where debris flows had occurred more often than other locations, and almost once every year in the past 18 years.

Given the environmental conditions at the mountain areas in Taiwan, it is more desired to deploy the monitoring instrumentation at the potentially streams, especially at the upper sections of them. But it usually encounters problems, like the power supply and communication interruption, when trying to install sensors at distant mountain areas. In Taiwan, the debris flow monitoring system had been built by Taiwan government, the Soil and Water Conservation Bureau (SWCB), by designing and applying different types of monitoring stations (Wang et al., 2011; Huang et al., 2013). Permanent and





automatic monitoring devices were installed on sites to collect data about debris flows. Therefore, the available data collected from the Shenmu Debris Flow Monitoring Station during the period of 2004 to 2021 were used for this work.

The studies about debris-flow characteristics started from Japan and China (Marchi et al., 2021). A review by Hürlimann et al. (2019) has addressed the collected information from nine monitoring sties and discussed the achievements and the types

of monitoring systems. The monitoring site conditions, local features of Shenmu area, and debris flow events were used as an important monitoring case in many studies (Huang et al, 2013; Huang et al., 2016; Lee at al, 2017; Wei et al, 2018; Hürlimann et al., 2019).

A dataset of debris flows recorded in the Shenmu area between 2004 to 2021 is presented in this study. The event data were

collected from various available sources. The data prepared in this study include date of debris flow events, time of debris flow occurrence (whenever available), the triggering rainfalls (rainfall intensity, accumulated rainfall and duration), and a brief review of soil moisture and vibrational signal graphs (some events). The rainfall data obtained and used in this study are from the Shenmu Village rainfall station ($23°31.9645'$N, $120°50.62'$E) maintained by the Central Weather Bureau (CWB) and from the Shenmu debris flow monitoring station ($23°31.6938'$N, $120°51.3927'$E) maintained by the Soil and Water

Conservation Bureau (SWCB). These data were collected from the instrumentation installed in the study area and were used to describe the important characteristics of debris flows in the Shenmu area. The video cameras which are available online are used to estimate the flow speeds of several events.

Analysis of debris-flow rainfall thresholds is also discussed from different approaches. The change of rainfall thresholds

used for debris flow early warning by the local government, and the characteristics of maximum hourly rainfall, effective accumulated rainfall, and rainfall intensity-duration (I-D) of debris flow events in Shenmu are described and compared here.

## 2 Study area and monitoring system

The study area is at the Shemu Village, the location of three highly potential debris-flow torrents. The three debris-flow

torrents, Aiyuzi Stream (DF226), Huosa Stream (DF227), and Chusuei Stream (DF199), join into Heshe Stream around the entrance of the village and the entire basic area is about 72.2 km$^2$ (Huang et al., 2013; Hürlimann et al., 2019). These streams belong to the watershed of Chen-Yu-Lan River in the central part of Taiwan. The terrain and landslide areas of streams and the basic info are shown in Fig. 1and Table 1, respectively. In the study area, the debris flows commonly occurs at the Aiyuzi Stream due to its shorter length and larger landslide areas located in its upstream since Typhoon Morakot in 2009

(Huang et al., 2013). The slope angles in the upper stream areas of the debris flows are between 30º and 50º and especially the average slope in the Aiyuzi Stream watershed is about 39.3º (Wei et al., 2018), based on the fact that over 75% of the

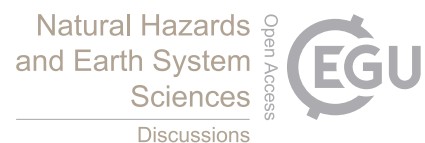

Aiyuzi Stream watershed area has slopes greater than 55% (Huang et al, 2007). A great percentage of the watershed was affected by significant landslides (Fig. 1). With the observation from the sites and from the videos of debris flows in Aiyuzi Stream, the debris flows in the Shenmu area are classified as granular flows. In many field investigations, boulders are often

1-2 m in diameter, and boulders with a diameter of more than 4-5 m can be seen in the accumulation of debris along streams.

Among the three debris-flow torrents in the study area, the Aiyuzi Stream is the focus of this study. Aiyuzi Stream is the stream in the upper stream of the Chen-Yu-Lan River catchment area. The total length of this stream is about 3.731 km, the watershed area is 405.02 hectares, and the topographical elevation is between 1,200 to 2,500 m. According to Chen et al.

(2012), the landslide ratio (landslide projected area/total catchment area) of this catchment area is about 12% ~ 34.2% (during 1996 to 2009), and there is a trend of increasing year by year.

Based on the rainfall data (from June 1987 to February 2017) from the Shenmu Village Station of the Taiwan's Central Weather Bureau (CWB), the average annual rainfall in the Aiyuzi Stream watershed area is about 3,054.7 mm, of which the

average total accumulated rainfall in the rainy season (from April to October each year) is 2,644.5 mm (Wei et al., 2018). Because of the abundant rains and landslide-induced debris at the upstream, with steep slopes (about 39.3$^\circ$ in average) at the region, debris flows often occurred in the Aiyuzi Stream during heavy rains and typhoons.

Since the ChiChi Earthquake in 1999, the environmental conditions in the mountain areas at the central Taiwan had become

vulnerable to heavy rainfalls and typhoons. Rainfall-induced shallow landslides occurred more often than ever, resulting in abundant soil, rocks, and debris accumulated in the streams. Debris flow disasters in 2000 had caused general public's attention since then. To protect the people living at the debris-flow prone areas, the Taiwan government started to establish the debris flow monitoring stations in 2002 and later had developed mobile monitoring stations and portable units to meet different needs. More and more observation data were obtained as the debris flow almost occurred annually from 2004 to

2009. In the study area, the debris flow occurred in 2004 was the first officially observed event with available monitoring data.

The Shenmu debris flow monitoring station was established in 2002 and is installed at 1,187 m asl at the confluence of three

potential debris flow torrents (Fig. 2). Different sensors are installed on the site of the monitoring station, to observe and

collect data for debris-flow warning and disaster repsonses. The monitoring options installed in the Shenmu aera includes direct and indirect measures for debris flows. The direct option refers to the devices that can capture, issuing warnings, and measure the status of debris flow when occurred. These devices used in Shenmu are wire sensors and geophones. The wire sensor simply issues a signal "broken" when a debris flow passing through and breaking the wire over the channel. The geophone detects the ground surface vibration, and the debris flow is recognized when a peak from the vibration records is





evident (Huang et al, 2016; Huang et al, 2007; Huang and Chen, 2022). The signals from the wire sensors and geophones
indicate the occurrence of debris flow and the timing when a debris flow is running through the locations of these
instruments. Other monitoring options, including rain gauges, soil moisture sensors, CCD camera, and flow speed meter are
used as supporting devices to help confirm the occurrence of debris flows. False alarms are inevitable during the debris flow
monitoring and cross-check is necessary and important.


Among the monitoring data, rainfall is the most used and important information for debris flow warning and analysis. The
signal of ground surface vibration is another considerably useful factor for indicating the occurrence of debris flow (Huang
et al., 2017; Wei et al, 2018). The change of water content at the underground soil is detected by the soil moisture sensors,
and the data are useful for potential estimation of shallow landslides. Figure 2 shows the detailed layout of the Shenmu

monitoring station, including rain gauges (TK-1, Takeda Keiki Co.), geophones (GS-20 DX, Geospace), wire sensors, soil
moisture sensors, and video cameras (PTZ camera). Similar to the geophones, two broadband seismic sensors (Yardbird DF-
2, Academia Sinica Taiwan) are also installed along the Aiyuzi Stream (Huang et al., 2017).

The Typhoon Herb in 1996 had caused debris flows in the channel of Chusuei Stream. Frequent debris flows occurred in the

Shenmu area after 1996. In July 2001, with the heavy rainfalls, Typhoon Toraji had caused sever casualties and property loss
to the people in the Shenmu area. Therefore, SWCB started to build the debris flow monitoring station in Shenmu since
2002. From that time, the local monitoring system has operated to continuously collect the observation data of debris flows
in the study area. Table 2 shows the observed event list of Shenmu area.

The events in Table 2 implies that the debris flows occurred almost every year during 2009 to 2014. After the extreme
rainfall event (1,550 mm in three days) of Typhoon Morakot in 2009, all the observed debris flows occurred at the Aiyuzi
Stream. The possible explanation to this condition is the shorter length and relatively more landslides at the upper-stream
areas of Aiyuzi Stream after 2009, when compared with the other two streams (Table 1). With abundant debris source at the
upstream sections and heavy rainfalls, these factors contribute to the frequent occurrence of debris flow in the Aiyuzi

Stream. After 2014 until 2021, there is only one record of debris flow in Shenmu (Table 2). It seems that the occurrence
frequency has reduced from once per year during 2009 to 2014, to very few in the past 7 years (from 2015 to 2021).

The main cause of the frequent debris flow in Shenmu was partially from the abundant debris source because of the
landslides. The landslide increments in Table 3 were estimated based on the satellite image processing. When considering

the landslide areas at the mid-to-upper stream sections, it has been noticed that the variation of landslide increments was
quite different before and after 2009. In August 2009, Typhoon Morakot had caused severe landslides at the study areas and
Fig. 3 shows the abrupt jump of landslide increments in 2010. The increment ratio of landslide at Aiyuzi Stream was higher
than that at the other two streams. Given the conditions of smaller catchment area and higher landslide increment ratio,





debris flows were expected to be easier to occur at the Aiyuzi Stream because of its abundant debris source at the upstream
sections. The number of post-2009 debris flow events confirmed the inference about the frequent debris flows at the Aiyuzi
Stream.

As stated above, from 1999 to 2009, the environment of the study area had tended to reach a balance after the ChiChi
Earthquake at central Taiwan. The landslide increment had eased during this period. After 2009, the environmental
conditions had significantly changed due to the Typhoon Morakot in 2009. More landslide areas had resulted in debris flows
every year during 2009 to 2014. Until now, the Shenmu is still under the threat of debris flow during typhoons and heavy
rainfalls.

According to the available satellite images from 2009 and 2014 to 2017, the mid- and downstream section widths of the
three streams had changed marginally (Table 4). The widest downstream channel is at the Huosa Stream, about 121 m, and
the width of Heshe Stream, the stream of confluence, is about 132 m since 2009. It is noted that the debris flows didn't
change the stream channel width too much but had accumulated abundant sediment and debris during and after the debris
flows (Fig. 5).

**3 Characteristics of debris flow at the study area**

**3.1 Rainfall Thresholds- RTI**

The current rainfall warning threshold of Shenmu area was developed by Jan and Lee (2004) and an index, Rainfall
Triggering Index (RTI), which is the product of hourly rainfall ($I$) and effective accumulated rainfall ($R_t$) was used to
determine the rainfall level of debris flow warning. For a given rainfall event, the value $R_t$ reflects the influence of
antecedent rains of a 7-day period and is calculated following Eq. (1). Fig. 6 illustrates the definitions of a rainfall event. The
Eq. (1) shows the expression of $R_t$, where $R_o$ is the current daily rainfall of an event, $R_i$ is the daily rainfall of previous i-th
day before current event and $\alpha$ is the weighting factor applied to antecedent daily rainfall. The weighting factor changed
normally between 0.5 to 0.8 based on the rainfall patterns and $\alpha$ of 0.7 is applied to the current debris flow early warning in
Taiwan.


$$R_t = R_0 + \sum_{i=1}^{7} \alpha^i \times R_i \tag{1}$$





The value of $R_t$ is divided into nine levels, from 200 mm to 600 mm with intervals of 50 mm, to represent the rainfall thresholds at different locations. Based on the debris-flow events, the rainfall threshold of the study area was 250 mm during

2000 to 2020 and is 300 mm since 2021.

### 3.2 Rainfall Thresholds- $I_{max}$, $R_{24}$, $R_t$

According to Huang et al. (2013), the case history of debris flows and heavy rainfalls in the Shenmu area implies rainfall thresholds of the maximum hourly rainfall ($I_{max}$) of 8 mm, and the effective accumulated rainfall ($R_t$) of 67 mm, before 2012. By adding additional and available rainfall data, Table 5 shows the events during 2014 to 2017. In addition to the effective

accumulated rainfall, the accumulated rainfall of 24 hour ($R_{24}$) before the occurrence of a debris flow was considered in this study. For those events that no debris flow occurred, the $R_{24}$ was estimated by assuming the debris flow would occur at the time of $I_{max}$ during an event. Fig. 7 and Fig. 8 show that plots of $I_{max}$-$R_{24}$ and $I_{max}$-$R_t$ based on the data of Huang et al. (2013) and this study. The plots show a slightly different rainfall thresholds of $I_{max}$ of 9 mm and $R_t$ of 67.8 mm, and $R_{24}$ of 23 mm as another rainfall threshold. The cases of $I_{max} \geq 9$ mm and $R_{24} \geq 23$ mm, and $I_{max} \geq 9$ mm and $R_t \geq 67.8$ mm are 33 and 32

cases, respectively, from the collected data. Among them, there are 10 debris flows, implying that the case percentage of debris flow is 30.3% and 31.3%, respectively. Another sets of rainfall thresholds are proposed as $I_{max} \geq 16.5$ mm and $R_{24} \geq$ 51.5 mm, and $I_{max} \geq 16.5$ mm and $R_t \geq 75.2$ mm, which results in the debris-flow case percentages of 40.0% and 38.1%, respectively. The abovementioned approach represents the debris-flow warning margins of rainfall in terms of $I_{max}$, $R_{24}$, and $R_t$, as shown in the plots (the orange and red dash-lines). These numbers may be not high enough to be used for practical

prediction of debris-flow occurrence. However, based on the case history, these rainfall thresholds ($I_{max}$ and $R_t$), as well as the RTI threshold of 300 mm, are considered reasonable as check points for debris flow disaster monitoring in the Shenmu area.

### 3.3 Rainfall thresholds- I-D curve

In addition to the rainfall thresholds of RTI factor, the statistics of rainfall data obtained about the occurrence of debris flows

shows additional characteristic relation of rainfall intensity ($I$) and duration ($D$). The records of rainfall data were collected from the rain gauges installed near and at the site of debris flow monitoring. The I-D relation has been studied and considered as the basis of defining rainfall thresholds for debris flows. Early research reveals the potential I-D curves from the global event data (Caine, 1980; Guzzetti et al., 2008). The I-D data of debris-flow events in Shenmu area is listed in Table 2 and the I-D curves are illustrated in the following Fig. 9 to Fig. 11.


Fig. 9 shows the intensity-duration plots of debris flows in Shenmu. The I-D data in this figure was estimated using the whole duration of each event and were obtained from the Shenmu Village Rainfall Station of Central Weather Bureau (CWB) and Shenmu Debris Flow Monitoring Station of Soil and Water Conservation Bureau (SWCB), respectively. The regression trends of data from both stations are similar with slight difference (dash-line of CWB and dash-dot line of SWCB,

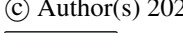


in Fig. 9), due to the locations where rainfall was measured. It was also noted that the trends (the slopes of regression curve) in Fig. 9 were not in consistent with those from previous research (Caine, 1980; Guzzetti et al., 2008). All the I-D data of Shemu fall above the global minimum rainfall I-D threshold (the blue line in the figure).

        Fig. 10 shows the I-D plots estimated during the period from the event start to one hour before the occurrence of debris
flows. It was noted that the trends of Fig. 10 were quite different to those in Fig. 9 and were considerably in consistent with those from previous research. Most of the data in Fig. 10 are within 48-hr duration. The trend line of SWCB in Fig. 10 is highly consistent with the one (green line) of 48-hr duration from the literature (Guzzetti et al., 2008), with similar slopes. The results from Fig. 10 implies that the duration hours used for estimating the rainfall I-D of each event should be counted based on the period from the event start to one hour before the occurrence of debris flow, rather than counting on the whole
event durations. In addition, the rainfall data from SWCB are more reasonable in analysis since the rainfall station of SWCB is closer to the locations of debris flow occurrence than that of CWB.

        Among the debris flow events in Shenmu, a catastrophic event of Typhoon Morakot in 2009 had caused extremely heavy rainfalls in the mountain areas, resulting in severe landslides, floods, and debris flows. Because of the significant impact of
Typhoon Morakot in 2009, the geological conditions and local environment of Shenmu had greatly changed. Different trends in the plots of rainfall data obtained before and after 2009 revealed the influence of severe weathers, as shown in Fig. 11. Both trends of post-2009 rainfall data show similar sloping as those from the literatures. Among these curves, the trend obtained from the rainfall data of SWCB station is highly in consistent with the curve of 48-hr duration from the literature. The post-2009 data regression curve of SWCB (i.e., $I = 16.01 \times D^{-0.238}$) reasonably represents the rainfall I-D of debris flows,
with the minimum boundary of 48-hr duration line ($I = 7.01 \times D^{-0.238}$) as reference. The hourly rainfall of 7 mm from the Fig. 11 is comparable with the proposed debris-flow rainfall threshold 9 mm of $I_{max}$ in this study. Both values can be considered as the warning triggers and used to define the response procedures for debris flow disaster management in the Shenmu area.

**3.4 Debris-flow flow velocity**

        The flow velocity was estimated by geophone data and CCD video clips in several events at the Aiyuzi Stream. Table 6
summarizes the flow velocity estimation, as well as the rainfall information before the occurrence of debris flows. It had been noted that the flow velocity of debris flow was quite higher before 2009, with the average velocity about 14.4 m/sec. After 2009, the average flow velocity of debris flow was much lower, about 4.5 m/sec, except the one obtained in the 1110 Heavy Rainfall in 2011, which was only about 1.8 m/sec. The possible explanation of the lower flow speed of debris flow at the Aiyuzi Stream after 2009 is the reduced slope at the mid- to downstream sections, due to the accumulation of debris from
events.





### 3.5 Soil moisture and vibrational signals

Regarding the debris-flow warning, the soil moisture obtained from the site provides another aspect of evaluation. The variation of soil moisture with time had been analysed with the rainfall records and the occurrence of debris flows (Tsai et al., 2014). However, the continuous measurement of soil moisture was not well carried out within the whole monitoring

period. The lack of this information had made the regression analysis of debris-flow occurrence and soil moisture not available. Another factor used for determination of debris-flow occurrence are the ground vibrational signals. These data are mainly used for research purpose and considered as additional information for debris-flow disaster response. (Huang et al., 2016; Huang et al., 2017). Previous study has shown that the characteristic frequency of the ground vibration signals generated byt the debris flows was between 10 and 40 Hz (Huang et al., 2016; Huang et al., 2017). The processed broadband

signals revealed the lower frequency of debris-flow fronts between 0 and 10 Hz and were captured earlier than those from the geophones (Huang et al., 2016). Figure 12 shows that the processed signal data (in terms of cumulative energy in the figure) had peaks when debris flows approached the monitoring station. The analysis results from studies indicate that the ground surface vibrational signals are potentially useful for debris-flow early earning.

From the recent research (Wei et al, 2018), it was found that the cumulative energy and the slope of cumulative energy in the 10-second interval (overlapping 50%) and in the 5-40 Hz frequency can be used as the interpretive indicators of the debris flow. However, the short time window between the time detected by sensors and the arrival of debris flows at the residential areas limits the capability of ground surface vibration signals in debris-flow early warning. Current research gives a promising indicator but is not applicable for disaster response.


### Conclusions

With examination of several aspects of the debris flow events in Shenmu area, the analysis led to the following conclusions.

The rainfall characteristics of debris flow in Shenmu mainly focus on the rainfall thresholds, in terms of maximum hourly

rainfall ($I_{max}$), 24-hour accumulated rainfall ($R_{24}$), and effective accumulated rainfall ($R_t$). The value of RTI-based effective accumulated rainfall has changed from 250 mm (during 2000 to 2020) to 300 mm (since 2021), due to the declining occurrence of debris flows in the area. Based on the case history in the Shenmu area, other proposed sets of rainfall thresholds are $I_{max}$, $R_{24}$, and $R_t$ are (9, 23, 67.8) and (16.5, 51.5, 75.2), all in mm. These values are considerably useful for debris flow disaster monitoring.


Another perspective of rainfall characteristics in Shenmu is the rainfall intensity and duration when debris flow occurred. Debris flows usually occur during the rainy season, especially during May to August of the year in this area. They are more



likely to occur during heavy rainfalls or intense typhoons. Precipitation thresholds of the initiation of debris flows are comparable to a global dataset. Compared to the global threshold of rainfall duration less than 48 hours (D<48), the regression curve of rainfall I-D in the study area has similar slope, indicating that shorter heavy rainfalls with average $I \geq 7.0$ mm h$^{-1}$ could trigger debris flows after 2009.

Based on the observed debris flow events, the average frequency of occurrence is about 1.11 times per year during the period from 2004 to 2021. Comparatively, the average frequency of debris flow occurrence is 1.83 times per year from 2004 to 2009 and is 0.75 times per year after 2009. The average frequencies of debris flow occurrence in Shenmu area implies that there was at least one debris flow occurred every year in the past 18 years. Also, it is noted that the occurrence frequency before 2009 is about twice the value after 2009, indicating that the impacts of extremely heavy rainfalls, the Typhoon Morakot in 2009, had significantly changed the local environment and the basin conditions at the upper stream sections. A lot of debris had been transported from the upper streams in 2009. Therefore, the debris source coming from the unstable shallow slopes along the upper stream sides may have reduced in a large amount after 2009. In fact, there is no debris flow observed in the study area since 2018. We expect that the amount of debris source is getting to a balance level in this area.

The reducing trend of debris flow occurrence after 2009 is reasonably due to the less landslides at the upper stream areas and fewer events of heavy rainfalls. The debris source has reached to a balance after 10 years since the Typhoon Morakot in 2009. Another possible evidence of fewer occurrence of debris flow is the flow speed of debris flows. The observed flow speed of debris flow from available data was abut 14.3 m/sec before 2009 and had become to a much lower value of 4.46 m/sec after 2009. The reasonable explanation of the slower flow speed after 2009 is due to the gentler slopes at the mid-to-downstream sections after years of debris settlement and less intensity of rainfall events in the past decade.

The monitoring results of past events shows that the peak values and rising energy from the ground surface vibrational signals were detected a few minutes earlier before the debris-flow's arrival at the monitoring devices. The characteristic frequency of signals was between 0 and 40 Hz and the cumulative energy and the slope of cumulative energy in the 10-second interval (overlapping 50%) can be used as the interpretive indicators of the debris flow. However, the limitation of short evacuation time window needs to be improved, either by more data analysis or different monitoring layout on the site, before being used as the indicator of debris-flow occurrence in the future disaster response.

The dataset prepared in this study is useful for further analysis regarding the mechanism and characteristics of debris flows in Taiwan, as well as advantageous to the global debris-flow dataset. It should be mentioned that the time series of rainfalls recorded by the rain gauges in the Shenmu area for the whole monitoring period is not completely available. Another problem is that the lack of systematic topographic surveys had restricted the estimation of sediment deposited along the stream channels in the Shenmu area, resulting in the difficulty of evaluating the magnitude of debris flows. The previous





issues could be resolved by conducting regular on-site surveys and with the help of advanced monitoring technology in the future.

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


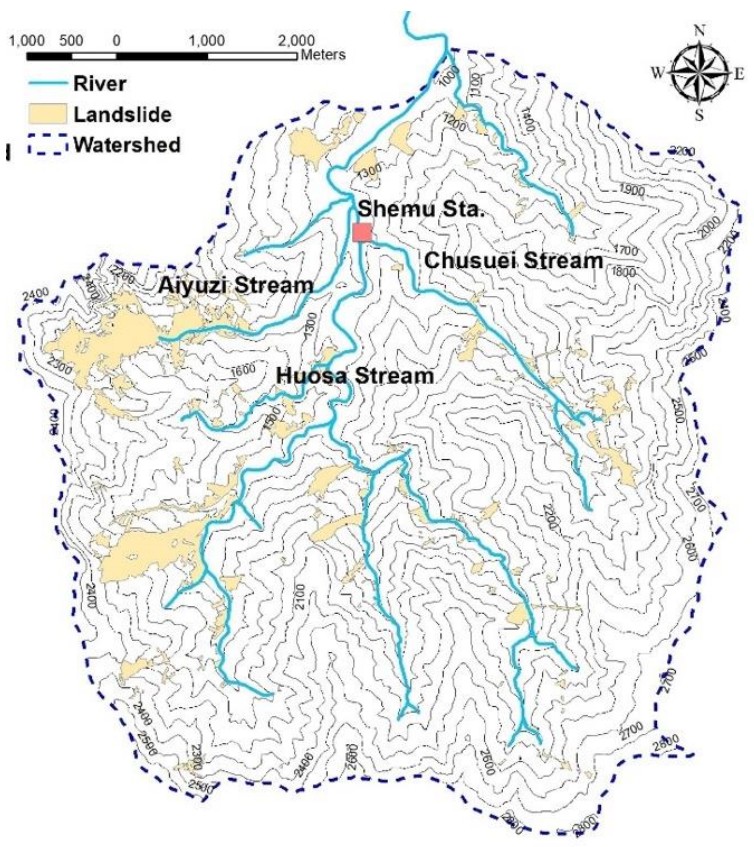

**Figure 1: The maps of terrain and the landslide areas (after 2009) of Shenmu area. (adapted from Huang et al., 2019)**



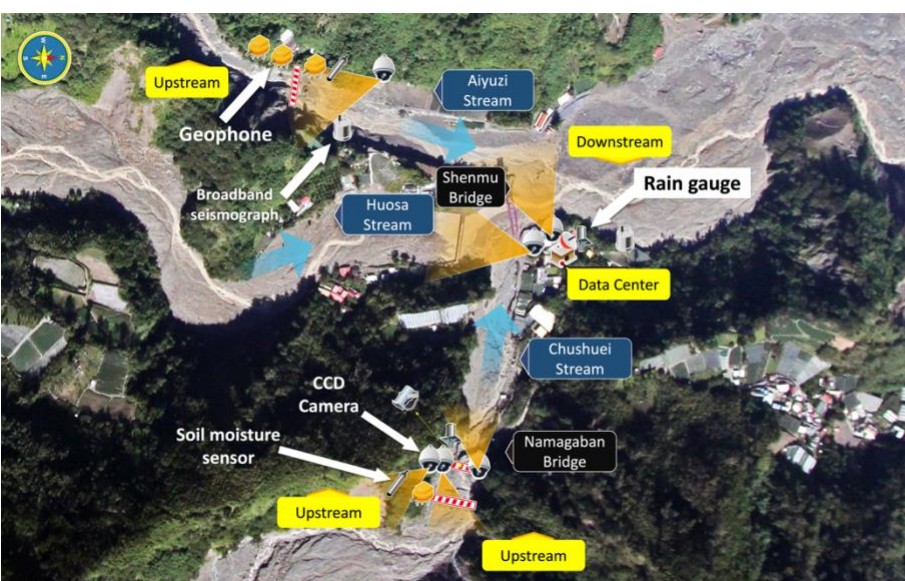

**Figure 2: The layout of sensors installed at the Shenmu monitoring station. (base map credit: GIS Research Center, Feng Chia University, Taiwan).**


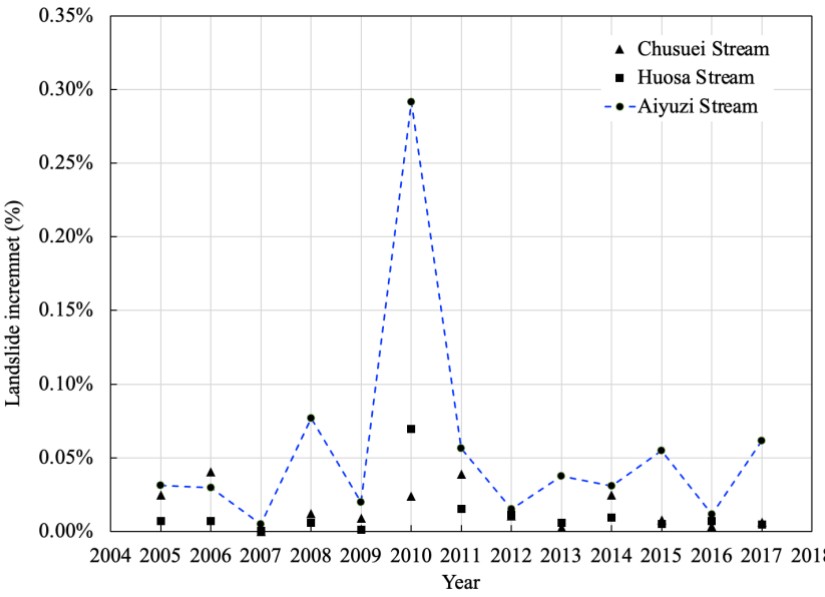

**Figure 3: Landslides increment ratio of the study area.**





**Figure 4: The landslide increments of (a) 2009 (b) 2010 (c) 2011 (d) 2017.**





Figure 5. The photos of the site near the mouth of Aiyuzi Stream (a) Aug. 12, 2009 (b) July 19, 2011 (c) Sep. 26, 2015 (d) June 9, 2017. (Photo credits: GIS Research Center, Feng Chia University, Taiwan).




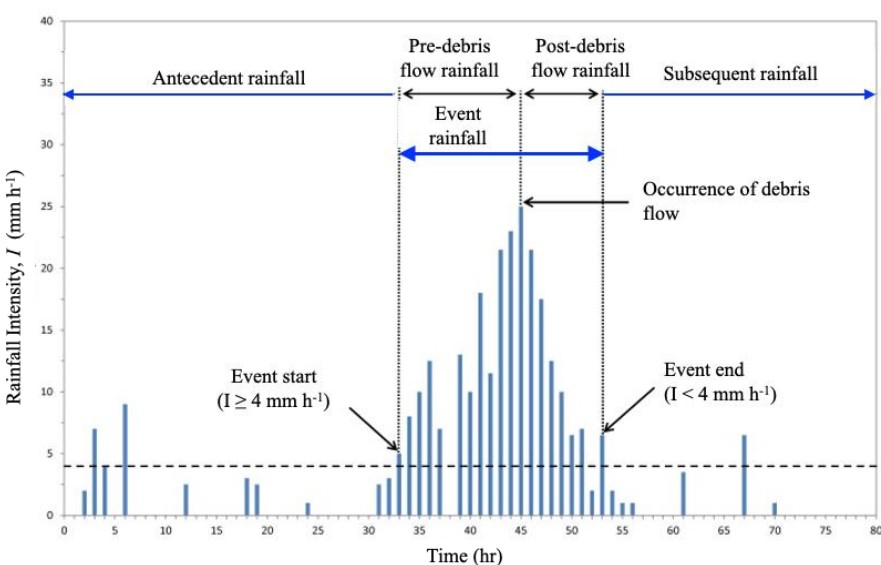

**Figure 6: Schematic diagram of the definition of a rainfall event.**


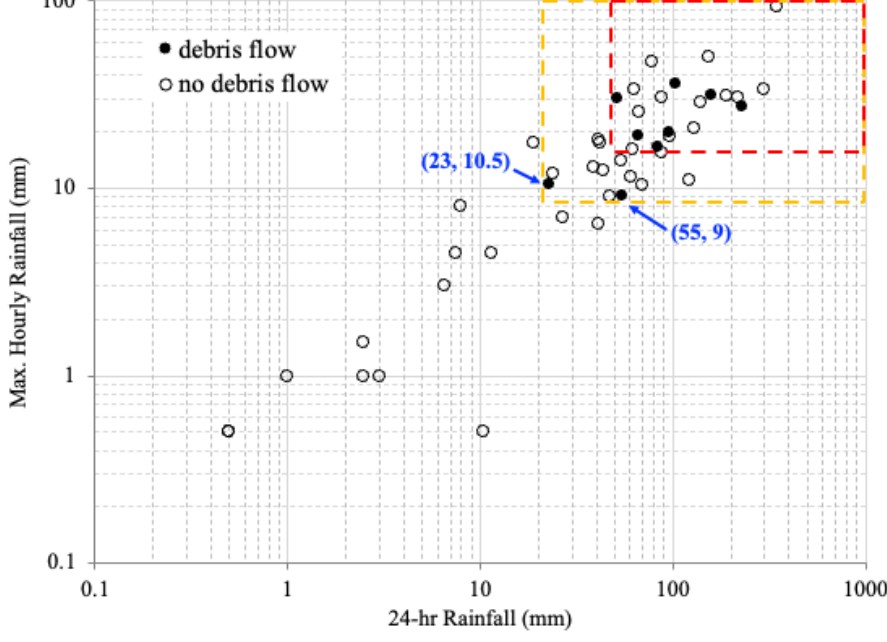

**Figure 7: The max. hourly rainfall and 24-hr rainfall of events in the Shenmu area.**




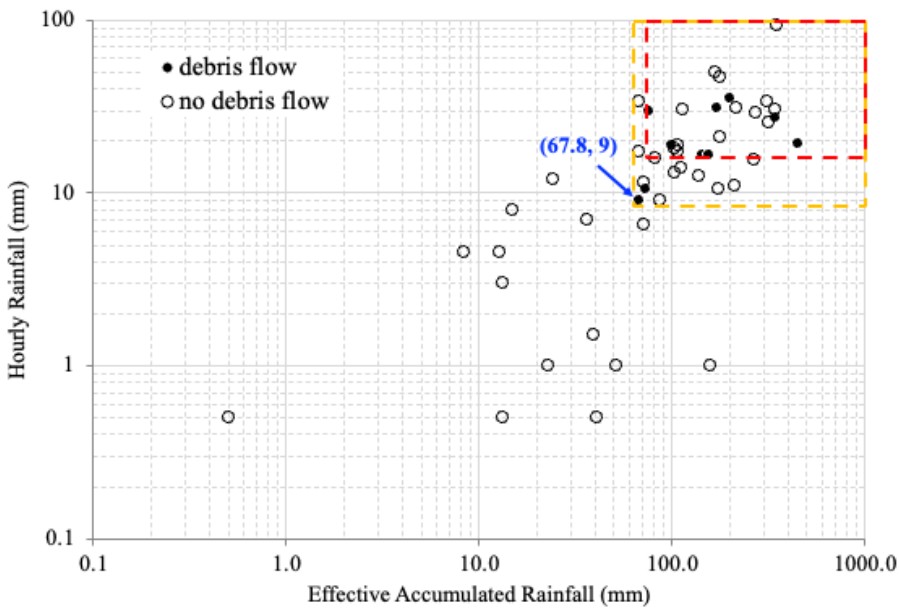


**Figure 8: The max. hourly rainfall and effective accumulated rainfall of events in the Shenmu area.**

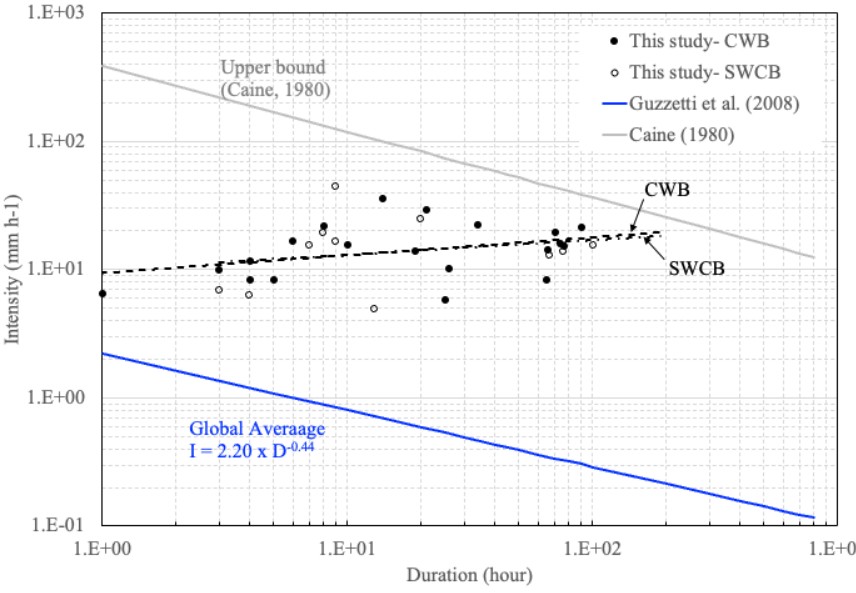

**Figure 9: The rainfall intensity-duration conditions of debris flow in Shenmu.**


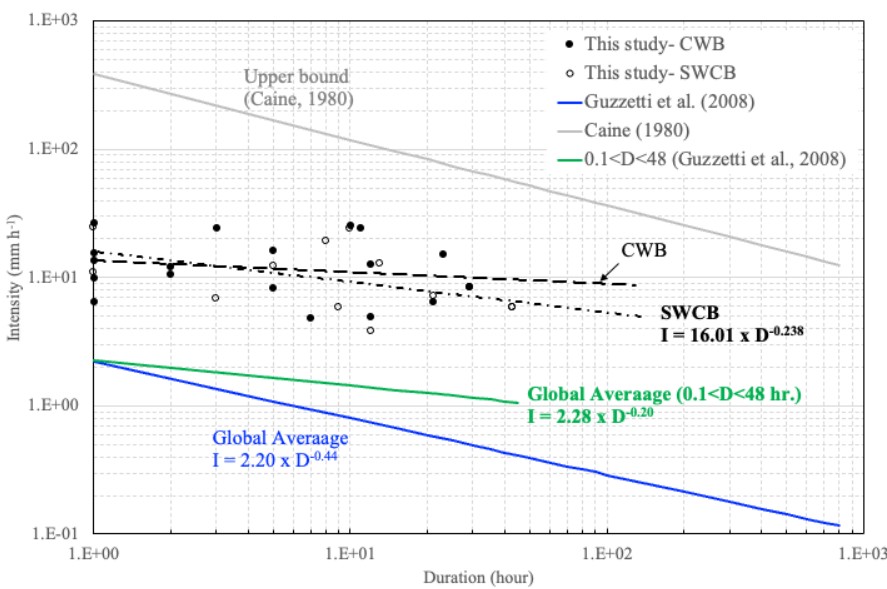


**Figure 10: The I-D conditions 1 hour before the occurrence of debris flows in Shenmu.**

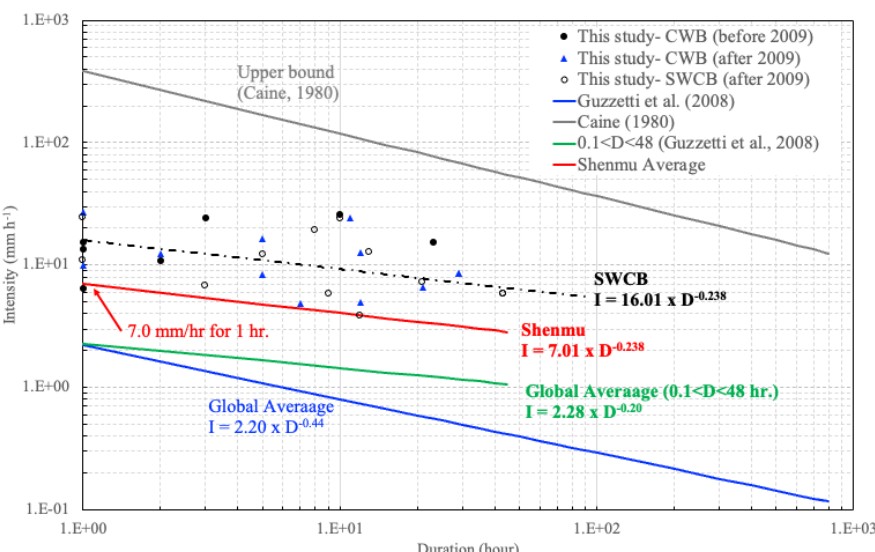

**Figure 11: The I-D conditions of debris flows in Shenmu before and after 2009.**




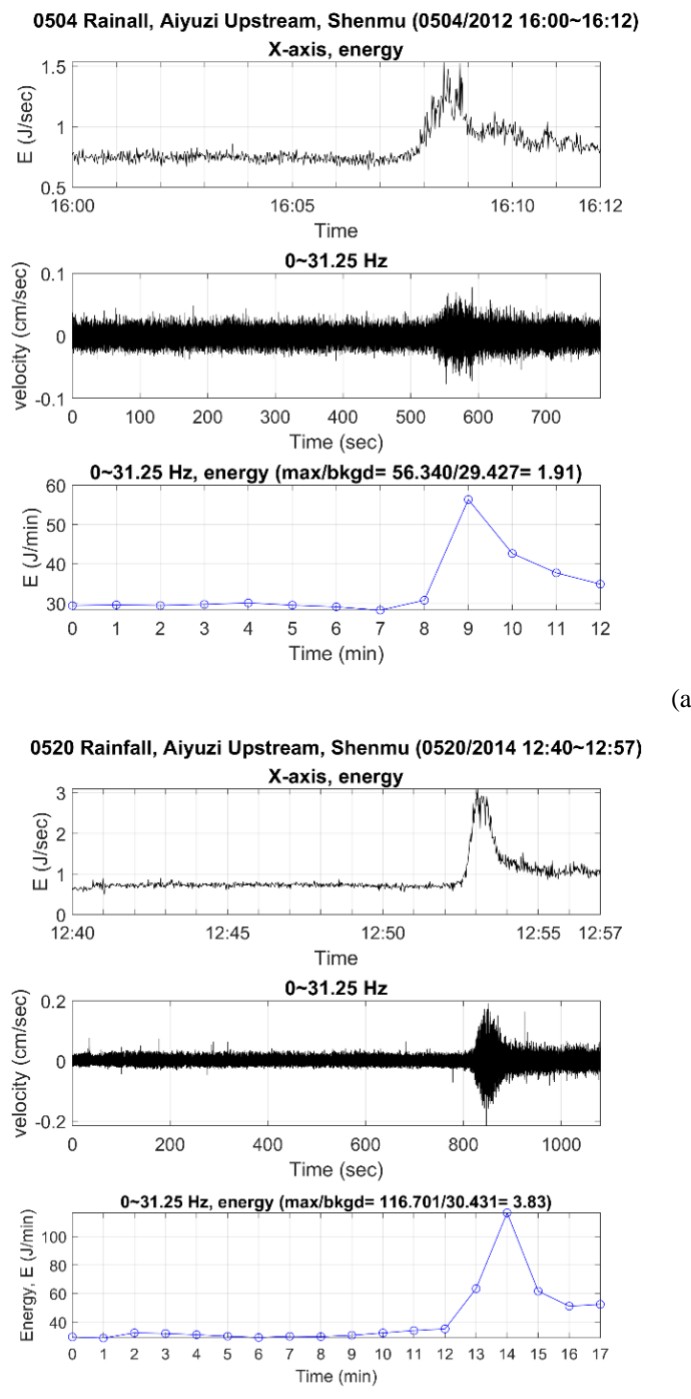

(a)

(b)

Figure 12: Energy of vibrational signals (a) 0504 Rainfall in 2012 (b) 0520 Rainfall in 2014 (Huang et al., 2019).






**Table 1: The landslide area in Shenmu after 2009 (Huang, et al., 2013)**

| Debris Flow No. | Stream | Length (km) | Catchment Area (ha) | Landslide Area (ha) |
|---|---|---|---|---|
| DF199 | Chusuei Stream | 7.16 | 861.56 | 33.29 |
| DF227 | Huosa Stream | 17.66 | 2,620 | 149.32 |
| DF226 | Aiyuzi Stream | 3.30 | 400.64 | 99.85 |


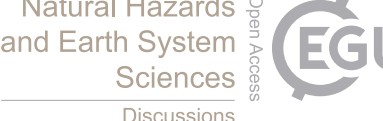

**Table 2: Debris flow events in the Shenmu area.**

| No | Date (Y/M/D) | Event | Location (stream) | Time of occur. | Warning (Y/N) | Hazard Type | Shenmu Rainfall Sta. (CWB) | | | | | Shenmu Monitoring Sta. (SWCB) | | | | |
|---|---|---|---|---|---|---|---|---|---|---|---|---|---|---|---|---|
| | | | | | | | Date/Time | $I_1$ | $D_1$ | $I_2$ | $D_2$ | Date/Time | $I_1$ | $D_1$ | $I_2$ | $D_2$ |
| 1 | 2004/5/20 | 0520 Heavy Rainfall | Aiyuzi | 14:53 | - | DF | 5/20/13:00~16:00 | 11.67 | 4 | 10.8 | 2.0 | 5/20/13:00~16:00 | - | - | - | - |
| 2 | 2004/5/21 | 0521 Heavy Rainfall | Aiyuzi | 16:08 | - | DF | 5/21/16:00~21:00 | 16.75 | 6 | 13.5 | 1 | 5/21/16:00~21:00 | - | - | - | - |
| 3 | 2004/5/29 | 0529 Heavy Rainfall | Aiyuzi | 16:19 | - | DF | 5/29/16:00~17:00 | 6.50 | 1 | 6.5 | 1 | 5/29/16:00~17:00 | - | - | - | - |
| 4 | 2004/6/11 | - | Aiyuzi | 16:42 | - | DF | 6/11/16:00~19:00 | 8.38 | 4 | 15.5 | 1 | 6/11/16:00~19:00 | - | - | - | - |
| 5 | 2004/7/02 | Typhoon Mindulle | Aiyuzi | 09:16 16:41 | - | DF | 7/2/08:00~7/5/6:00 | 19.73 | 71 | 24.3 25.7 | 3 10 | 7/2/08:00~7/5/6:00 | - | - | - | - |
| 6 | 2006/6/09 | 0609 Rainfall | Chusuei Aiyuzi | 08:32 (at Aiyuzi) | Y | DF | 6/8/11:00~6/11/12:00 | 15.85 | 74 | 15.3 | 23 | 6/8/11:00~6/11/12:00 | - | - | - | - |
| 7 | 2007/8/13 | 0809 Rainfall | Chusuei | - | - | DF, FLD | 8/12/12:00~8/13/13:00 | 10.15 | 26 | - | - | 8/12 12:00~8/13 13:00 | - | - | - | - |
| 8 | 2007/8/18 | Typhoon Sepat | Chusuei | - | - | DF, FLD | 8/18/00:00~8/20/17:00 | 8.38 | 65 | - | - | 8/18/00:00~8/20/17:00 | - | - | - | - |
| 9 | 2007/10/06 | Typhoon Krosa | Aiyuzi | - | - | DF | 10/6/09:00~10/7/19:00 | 22.53 | 34 | | | 10/6/09:00~10/7/19:00 | - | - | - | - |
| 10 | 2008/7/17 | Typhoon Kalmaegi | Chusuei, Aiyuzi | - | - | DF, FLD | 7/17/21:00~7/18/10:00 | 36.07 | 14 | | | 7/17/20:00~7/18/04:00 | 44.06 | 9 | - | - |
| 11 | 2009/8/08 | Typhoon Morakot | Chusuei Huosa Aiyuzi | at Aiyuzi: 04:39 | Y (04:41, 8/8) | DF, LS | 8/7/1:00~8/10/18:00 | 21.37 | 90 | 8.55 | 29 | 8/6/11:00~8/10/15:00 | 15.58 | 101 | 5.88 | 43 |
| 12 | 2011/7/13 | 0713 Heavy Rainfall | Aiyuzi | 14:33 | N | DF | 7/13/15:00~17:00 | 10.00 | 3 | 10.00 | 1 | 7/13/15:00~18:00 | 6.25 | 4 | 11.00 | 1 |
| 13 | 2011/7/19 | 0719 Heavy Rainfall | Aiyuzi | 03:19 | N | DF | 7/18/23:00~7/19/17:00 | 13.87 | 19 | 16.50 | 5 | 7/18/23:00~7/19/05:00 | 15.43 | 7 | 12.20 | 5 |
| 14 | 2011/11/10 | 1110 Heavy Rainfall | Aiyuzi | 13:29 | N | DF | 11/10/2:00~11/11/2:00 | 5.76 | 25 | 4.96 | 12 | 11/10/2:00~11/10/14:00 | 4.92 | 13 | 3.88 | 12 |
| 15 | 2012/5/04 | 0504 Heavy Rainfall | Aiyuzi | 15:56 | N | DF | 5/4 12:00~16:00 | 8.30 | 5 | 8.30 | 5 | 5/4/14:00~16:00 | 6.83 | 3 | 6.83 | 3 |
| 16 | 2012/6/10 | 0610 Rainfall | Aiyuzi | 10:34 15:14 | Y (20:33, 6/10) | DF | 6/10 4:00~6/12 21:00 | 14.38 | 66 | 4.86 12.75 | 7 12 | 6/10/3:00~6/12/21:00 | 12.88 | 67 | 5.83 12.81 | 9 13 |
| 17 | 2013/5/17 | 0517 Heavy Rainfall | Aiyuzi | 07:02, 5/19 | Y (07:02, 5/19) | DF | 5/19 6:00~13:00 | 21.75 | 8 | 12.25 | 2 | 5/19/6:00~13:00 | 19.44 | 8 | 19.44 | 8 |
| 18 | 2013/7/11 | Typhoon Soulik | Aiyuzi | 06:54, 7/13 | Y (06:44, 7/13) | DF | 7/12 20:00~7/13 16:00 | 29.40 | 21 | 24.36 | 11 | 7/12/22:00~7/13/17:00 | 24.55 | 20 | 24.10 | 10 |
| 19 | 2014/5/20 | 0520 Heavy Rainfall | Aiyuzi | 12:53 | N | DF | 5/20 13:00~22:00 | 15.75 | 10 | 27 | 1 | 5/20/13:00~21:00 | 16.39 | 9 | 24.5 | 1 |
| 20 | 2017/6/01 | 0601 Heavy Rainfall | Aiyuzi | 11:40, 6/02 | Y (13:44, 6/02) | DF | 6/1 15:00~6/4 19:00 | 15.40 | 77 | 6.52 | 21 | 6/1/16:00~6/4 19:00 | 13.86 | 76 | 7.21 | 21 |

$I_1$, $I_2$: the rainfall intensity (mm hr$^{-1}$) of the whole event and of the duration between the event start to one hour before the occurrence of debris flow, respectively; $D_1$, $D_2$: the rainfall duration (hour) of the whole event duration and of the duration between event start to one hour before the occurrence of debris flow, respectively; DF: debris flow; FLD: flood; LS: landslide. -: not available.





**Table 3.** Landslide increments (m$^2$) from 2005 to 2017 in Shenmu area.

| Year | Chusuei Stream | Huosa Stream | Aiyuzi Stream | Year | Chusuei Stream | Huosa Stream | Aiyuzi Stream |
|------|------|------|------|------|------|------|------|
| 2005 | 210,196 | 186,560 | 117,238 | 2012 | 91,007 | 293,469 | 56,752 |
| 2006 | 342,699 | 179,173 | 111,793 | 2013 | 22,853 | 154,012 | 141,260 |
| 2007 | 1,043 | 14,552 | 17,634 | 2014 | 210,291 | 245,631 | 115,568 |
| 2008 | 101,531 | 153,507 | 288,463 | 2015 | 65,171 | 127,627 | 205,958 |
| 2009 | 76,642 | 27,396 | 74,971 | 2016 | 31,272 | 188,270 | 43,086 |
| 2010 | 201,523 | 1,807,496 | **1,093,685** | 2017 | 51,913 | 120,809 | 230,972 |
| 2011 | 329,184 | 398,638 | 212,154 | | | | |

**Table 4. The channel width (m) of mid- to downstream.**

| Year \ Stream | Chusuei | Huosa | Aiyuzi | Heshe* (the confluence) |
|------|------|------|------|------|
| 2009 | 76.14 | 121.84 | 71.75 | 132.35 |
| 2014 | 74.88 | 121.64 | 71.48 | 132.81 |
| 2015 | 74.70 | 122.53 | 71.11 | 132.40 |
| 2016 | 75.56 | 121.83 | 71.42 | 132.89 |
| 2017 | 75.83 | 121.65 | 70.99 | 132.80 |

* The channel width (m) measured at about 200 meters downstream to the confluence of the three potential debris flow torrents.






**Table 5. Case history of rainfalls and typhoons in the Shenmu area from 2014 to 2017.**

| Event | Year | Mo | Day | $DF^*$ | $I_{max}$ (mm) | $R_{24}$ (mm) | $R_t$ (mm) |
|---|---|---|---|---|---|---|---|
| 0520 Heavy Rainfall | 2014 | 5 | 20 | Y | 35.5 | 103.5 | 199.15 |
| 0606 Heavy Rainfall | 2014 | 6 | 6 | N | 14 | 54 | 112.09 |
| Typhoon Hagibis | 2014 | 6 | 13 | N | 13 | 39 | 102.44 |
| Typhoon Matmo | 2014 | 7 | 23 | N | 30.5 | 219.5 | 341.79 |
| 0520 Heavy Rainfall | 2015 | 5 | 23 | N | 17.5 | 42 | 106.88 |
| Typhoon Chan-Hom | 2015 | 7 | 11 | N | 19 | 97.5 | 106.90 |
| Typhonn Soudelor | 2015 | 8 | 6 | N | 29 | 138.5 | 272.84 |
| 0611 Heavy Rainfall | 2016 | 6 | 11 | N | 18 | 41.5 | 103.30 |
| Typhoon Meranti | 2016 | 9 | 14 | N | 9 | 47 | 86.51 |
| Typhoon Malakas | 2016 | 9 | 16 | N | 47 | 77.5 | 179.08 |
| Typhoon Megi | 2016 | 9 | 27 | N | 31 | 189 | 217.50 |
| 0601 Heavy Rainfall | 2017 | 6 | 2 | Y | 19.5 | 97.5 | 451.78 |
| 0613 Heavy Rainfall | 2017 | 6 | 14 | N | 21 | 129.5 | 177.60 |
| 0613 Heavy Rainfall | 2017 | 6 | 15 | N | 11 | 122.5 | 211.32 |
| 0613 Heavy Rainfall | 2017 | 6 | 17 | N | 25.5 | 67.5 | 315.63 |
| Typhoon Nesat and Haitang | 2017 | 7 | 29 | N | 50 | 153 | 166.44 |
| Typhoon Nesat and Haitang | 2017 | 7 | 30 | N | 15.5 | 88 | 266.75 |

*$DF$: debris flow


**Table 6: The flow speed of debris flow in Aiyuzi Stream.**

| Event | Date | Flow Speed (m s$^{-1}$) |
|---|---|---|
| Typhoon Midule | July 2, 2005 | 13.3 |
| Typhoon Morakot | 4:36, Aug. 8, 2009 (first wave) | 13.0 |
|  | 16:57, Aug. 8, 2019 (second wave) | 17.0 |
| 0713 Heavy Rainfall | July 13, 2011 | 4.26 |
| 1110 Heavy Rainfall | Nov. 10, 2011 | 1.77 |
| 0610 Heavy Rainfall | June 10, 2012 | 4.26 |
| 0520 Heavy Rainfall | May 20, 2014 | 4.87 |