# Peer review of "Characteristics of debris flows recorded in the Shenmu area of central Taiwan between 2004 and 2021"

_Natural Hazards and Earth System Sciences, 2022_

## Author Comment (AC1)

**Natural Hazards and Earth System Sciences**

Dear Anonymous Reviewer,

Thank you for helping me to submit a revised draft of my manuscript titled **Characteristics of debris flows recorded in the Shenmu area of central Taiwan between 2004 and 2021** to *Natural Hazards and Earth System Sciences*. I appreciate the time and effort that you have dedicated to providing your valuable feedback on my manuscript. I am grateful to the reviewer's insightful comments on my paper. I have been able to incorporate changes to reflect most of the suggestions provided by the reviewers. I have highlighted the changes within the manuscript. After the revision, I believe that this study and its outcomes fulfil the scope of journal.

Here is a point-by-point response to the reviewers' comments and concerns.

**Comments from Reviewer RC1**

The author presented a valuable manuscript for a debris flow monitoring site. However, there are some issues needed to complete the manuscript.

- **Comment 1:** *The abstract should provide a general idea of this study, which includes the importance, the design, the process, and most important the finding from the data. The author should consider rewriting the abstract.*
  **Response:** Thank you for your suggestion. This study mainly focuses on the findings of rainfall factors, intensity (I), duration (D) and rainfall thresholds, that could contribute to the global debris-flow database. Another contribution of this study is to provide an overview of debris-flow monitoring system, available options for debris-flow early warning, and important characteristics about the debris flows in the Shenmu area Taiwan. The abstract of this study is revised in the updated manuscript (p. 1).

- **Comment 2:** *The introduction should provide a basic idea of the literature and methods, which is lacking in the present form.*
  **Response:** Thank you for your suggestion. This study mainly focuses on the findings of rainfall factors, intensity (I) and duration (D), that could contribute to the global debris-flow database. Another contribution of this study is to provide an overview of debris-flow monitoring system, available options for debris-flow early warning, and important characteristics about the debris flows in the Shenmu area, Taiwan. Therefore, the author reviewed researches about the study area, and provided description and important references regarding the debris-flow monitoring system, types of collected data, and recent

study results about the debris flows in the Shenmu area. The section of Introduction is rearranged and revised in the updated manuscript (p. 1).

- **Comment 3:** *Some names are different in Figure 1 and Figure 2.*
  **Response:** Thank you for your suggestion. The name of Shenmu Sta. is corrected in Figure 1, and the name of Chusuei Stream is corrected in Figure 2.

- **Comment 4:** *The author presents the debris period to 2021 but the data shown in tables and figures stopped in 2017.*
  **Response:** Thank you for your comment. The period of study is from 2004 to 2021, and there were no debris flows occurred after 2017. Therefore, the data in tables and figures obtained for debris-flow cases stopped in 2017. Additional description about the captured debris-flow events is added in the manuscript (see Line xxx to xxx).

- **Comment 5:** *In line 66, the author mentions that the Aiyuzi stream is the focus of this study but the manuscript keeps mentioning the other two streams.*
  **Response:** Thank you for your comment. There are three potential debris-flow torrents in the Shenmu area. To help readers better understand the local environment and the case history, the author included the description of all three streams in a few paragraphs, as addressed in the Section 2. Other than that, Aiyuzi Stream is the main focus of this study.

- **Comment 6:** *In lines 115 to 116, the rainfall event with 1550 mm is missing in Table 2.*
  **Response:** Thank you for your comment. The event of Typhoon of Morakot was included in Table 2, and the author tried to emphasize the extreme impacts of Typhoon Morakat by pointing out the three-day (Aug. 7 to Aug. 9) accumulated rainfall of this event. The value of 1,550 mm is corrected to 1,872 mm in the manuscript after revision (see Line xxx to xxx). Table 2 shows the average of hourly rainfall based on the durations.

- **Comment 7:** *The landslide increment seems incorrect when comparing Table 1 and Table 3.*
  **Response:** Thank you for your comment. Table 1 describes the total landslide areas after Typhoon Morakot in 2009, not the landslide increment. Table 3 shows the annual increment landslide areas, which area the difference between two consecutive years. The unit in Table 1 and Table are changed to m$^2$, to avoid confusion and for comparison.

- **Comment 8:** *In line 205, "Different trends" seems the same trend in Fig. 11.*

> **Response:** Thank you for your suggestion. The trend line of data from CWB (before 2009) is added to Fig. 11, to be used for comparison between the trends from the data before and after 2009 in the figure.

- **Comment 9:** *In line 216, the velocity before 2009 has only one data and is not the same as 14.4 m/sec in Table 6.*

  **Response:** Thank you for your suggestion. The average velocity is estimated before and after 2010, not 2009. The manuscript content has been correctee in the Section 3.4.

- **Comment 10:** *The soil moisture sensor can be withdrawn from the manuscript since there is no good or enough data to analyze.*

  **Response:** Thank you for your suggestion. Considering the monitoring system in the study area, the use of soil moisture should be addressed to provide a complete debris-flow warning approach in the study area. Although the soil moisture data may not be useful for debris-flow warning at present, it still can be a part of monitoring system and we continue to collect data for further analysis.

- **Comment 11:** *The findings from the geophone should have more descriptions and discussions with such valuable data.*

  **Response:** Thank you for your suggestion. Vibrational signals from geophones are promising for debris-warning based on the research. More detail can be found from the references, and recent study results from the author is added in the Section 3.5.

- **Comment 12:** *The first paragraph of the conclusions says the effective rainfall changed from 250 mm to 300 mm but the result is not shown in the previous part of this manuscript.*

  **Response:** Thank you for your suggestion. The description of rainfall thresholds is in the content of Section 3.1, more detail is added, and the paragraph is revised (see Line xxx to xxx).

- **Comment 13:** *The 3$^{rd}$ paragraph of the conclusions says that the frequency of debris flow is different with 1.83 and 0.75 times but cannot be seen in the previous part of this manuscript.*

  **Response:** Thank you for your suggestion. The analysis and results are added in the manuscript and the content of Section 2 is revised (see Line xxx to xxx).

**Additional clarifications**

**Natural Hazards and Earth System Sciences**

In addition to the above comments, all spelling and grammatical errors pointed out by the author and the editing supervisor have been corrected.

I look forward to hearing from you in due time regarding the submission and to respond to any further questions and comments you may have.

Sincerely,

Yi-Min Huang

Jan. 29, 2023

---

## Author Comment (AC2)

**Natural Hazards and Earth System Sciences**

Dear Dr. Chihping Kuo,

Thank you for helping me to submit a revised draft of my manuscript titled **Characteristics of debris flows recorded in the Shenmu area of central Taiwan between 2004 and 2021** to *Natural Hazards and Earth System Sciences*. I appreciate the time and effort that you have dedicated to providing your valuable feedback on my manuscript. I am grateful to the reviewer's insightful comments on my paper. I have been able to incorporate changes to reflect most of the suggestions provided by the reviewers. I have highlighted the changes within the manuscript. After the revision, I believe that this study and its outcomes fulfil the scope of journal.

Here is a point-by-point response to the reviewers' comments and concerns.

**Comments from Reviewer CC1**

This article presents a valuable evaluating procedure of threshold values to occur debris flow for a site. Some comments were proposed here for the author to revise or add in the article.

- **Comment 1:** *Line 7: the location name "Shebmu" is different from others in the article, please revise it.*
  **Response:** Thank you for your suggestion. The name of location is corrected as "Shenmu."

- **Comment 2:** *Line 8: the description "Total of 20 debris flows were observed in this time interval." seems to be 20 times of debris flow events occurred in this time interval but not 20 amounts. It is suggested to rewrite or clarify the description.*
  **Response:** Thank you for your suggestion. The content is modified as 20 times of debris-flow events, and the revised content is as follows.

  "**Abstract**. The data of debris-flow events between 2004 and 2021 in the Shenmu area Taiwan were presented and discussed in this paper. Total of 20 times of debris-flow events were observed in 18 years. Debris flows in the Shenmu area usually occurred in the Aiyuzi Stream during the rainy season, May to September, and about once per year between 2009 and 2017. The rainfall thresholds from the observed data were proposed as $I_{max}$, $R_{24}$, and $R_t$ of 9, 23, and 67.8 mm, respectively. The rainfall data also implied that the trend curves of intensity-duration (*I-D*) were different before and after 2009, which due to the extreme rainfall event, Typhoon Morakot in 2009. The I-D curve obtained from the post-2009 data was proposed as the baseline of debris-flow *I-D* relationship in the study area. The extreme rainfall event also influenced the flow speed (average 14.3 m/sec before 2010 and 4.46 m/sec after 2010) and the occurrence frequency of debris flow (1.83 times per year before 2009 and 0.75 times after 2009). Recent findings

indicated that the ground surface vibrational signals of debris flows were potentially useful for debris-flow early warning in terms of accumulated energy, and the characteristic frequency of debris flow in the study area was below 40 Hz. The dataset and the rainfall thresholds in this study permits the comparison with other monitored catchments and is advantageous to the global debris-flow dataset."

- **Comment 3:** *Line 42-45: two important rainfall data resources were proposed here. Please address them in Fig.1 or 2.*
  **Response:** Thank you for your suggestion. The rainfall stations are added and marked on Figure 1.

- **Comment 4:** *Line 54-56: is the abbreviation DF a name or index system for Debris Flow in Taiwan or this research?*
  **Response:** Thank you for your comment. The abbreviation DF stands for Debris Flow, and the index names of the potential debris flow streams in the study area are used in Taiwan.

- **Comment 5:** *Line 60-62: please unify the expression of slope, i.e. degree or percentage.*
  **Response:** Thank you for your suggestion. The slope in the manuscript is expressed in degree format.

- **Comment 6:** *The collection of data in this article is very complete. Does the author possible suggest an enough and proper period of monitoring for evaluating similar threshold values as characteristics of a site?*
  **Response:** Thank you for your comment. Based on the monitoring results in the study area, the author would suggest that the initial 5-year monitoring is necessary after the monitoring system started, and additional 5 years, resulting in total of a 10-year monitoring period, is favorable for data collection and analysis, especially when encountering unusual rainfall events and varied weather patterns.

- **Comment 7:** *The format of vertical and horizontal axis is suggested to be uniform in Fig.7 to 11.*
  **Response:** Thank you for your suggestion. The format of axes in Figure 7 and Figure 8 are changed to the same format as in Figure 9 to Figure 11.

**Additional clarifications**

**Natural Hazards and Earth System Sciences**

In addition to the above comments, all spelling and grammatical errors pointed out by the author and the editing supervisor have been corrected.

I look forward to hearing from you in due time regarding the submission and to respond to any further questions and comments you may have.

Sincerely,

Yi-Min Huang

Feb. 15, 2023